# Unplanned hospital visits after ambulatory surgical care

Tasce Bongiovanni[1]*, Craig Parzynski[2], Isuru Ranasinghe[3,4], Michael A. Steinman[5], Joseph S. Ross[2,6,7]

1 Department of Surgery, University of California San Francisco School of Medicine, San Francisco, CA, United States of America, 2 Center for Outcomes Research and Evaluation, Yale–New Haven Hospital, New Haven, Connecticut, United States of America, 3 Department of Cardiology, The Prince Charles Hospital, Brisbane, Australia, 4 School of Clinical Medicine, The University of Queensland, Brisbane, Australia, 5 Division of Geriatrics, University of California San Francisco School of Medicine and San Francisco VA Medical Center, San Francisco, CA, United States of America, 6 Section of General Internal Medicine, Yale University School of Medicine, New Haven, Connecticut, United States of America, 7 Department of Health Policy and Management, Yale University School of Public Health, New Haven, Connecticut, United States of America

* tasce.bongiovanni@ucsf.edu

**Data Availability Statement:** All HCUP files are available from the HCUP database, website: https://www.hcup-us.ahrq.gov/.

**Funding:** The authors received no specific funding for this work.

## Abstract

### Objectives

We sought to assess the rate of unplanned hospital visits among patients undergoing ambulatory surgery.

### Summary background data

The majority of surgeries performed in the United States now take place in outpatient settings. Post-discharge hospital visit rates have been shown to vary widely, suggesting variation in surgical or discharge care quality. Complicating efforts to address quality, most facilities and surgeons are unaware of their patients' hospital visits after surgery since patients may present to a different hospital.

### Methods

We used state-level, administrative data from the Agency for Healthcare Research and Quality's Healthcare Cost and Utilization Project from California to assess unplanned hospital visits after ambulatory surgery. To compare rates across centers, we determined the age, sex, and procedure-adjusted rates of hospital visits for each facility using 2-level, hierarchical, generalized linear models using methods similar to existing Centers for Medicare and Medicaid Services measures.

### Results

Among a total of 1,260,619 ambulatory same-day surgeries from 440 surgical facilities, the risk adjusted 30-day rate of unplanned hospital visits was 4.8%, with emergency department visits of 3.1% and hospital admissions of 1.7%. Several patient characteristics were

**Competing interests:** JSR has received research support through Yale University from Johnson and Johnson for other work not related to this study. The other authors have no competing interests to declare.

associated with increased risk of unplanned hospitals visits, including increased age, increased number of comorbidities (using the Elixhauser score), and type of procedure (p<0.001).

## Conclusions

The overall rate unplanned hospital visits within 30 days after same-day surgery is low but variable, suggesting a difference in the quality of care provided. Further, these rates are higher among specific patient populations and procedure types, suggesting areas for targeted improvement.

## Introduction

The majority of surgeries performed in the United States now take place in outpatient settings [1,2], with many performed as same-day surgeries at hospital outpatient departments (HOPDs) [1]. To meet this increasing demand for outpatient surgery, the number of ambulatory surgery centers (ASCs) has also grown quickly over the past 3 decades [3,4]. Further, lower prices charged by these freestanding ambulatory centers have stimulated interest among employers and insurers in encouraging employees and enrollees to select these facilities for their surgical care [5]. Procedures take less time at these centers, which keeps costs down and allows surgeons to keep up with demand [3]. While advances in anesthesia and the development of laparoscopic surgery have made same-day surgery a viable and safe option [6], the risk of adverse events remains, many of which are potentially preventable, such as uncontrolled pain, urinary retention, infection, bleeding, and venous thromboembolism. These events can result in unanticipated hospital visits post-discharge. Similarly, direct admissions for non-healthcare specific reasons after surgery, such as lack of transport home upon discharge, and other logistical issues, such as delayed start of surgery, are common causes of unanticipated yet preventable hospital admissions following same-day surgery.

As more surgery shifts to the same-day outpatient setting, there has been increased attention to unplanned hospital utilization following same-day surgery as an important patient-centered outcome of quality that reflects the patient's experience of adverse events after surgery [7–10]. National estimates of hospital visit rates following ambulatory surgery vary from 0.5% to 9.0%, based on the type of surgery, outcome measured (admissions alone or admissions and emergency department visits), and timeframe for measurement after surgery [7,9,11–18]. Post-discharge hospital visit rates have been shown to vary widely among HOPDs [9,18], suggesting variation in surgical and discharge care quality. Complicating efforts to better understand quality, most facilities and surgeons are unaware of their patients' hospital visits after surgery since patients may present to an ED at a different hospital [17].

Although recent studies have examined readmission rates following outpatient surgery, they have focused on just ASCs, assessed a short time period (7 days) of acute-care needs [18], or have been focused on only one specialty [19]. Further, as more outpatient surgeries are performed at HOPDs as part of a hospital based practice, instead of ASCs, it is increasingly important to be able to assess the variation in quality of these centers in comparison with each other. A deeper understanding of the variation in acute-care needs within 30 days of outpatient surgery is critical to improving quality. Prior measures looked only at Medicare patients; therefore it is unclear if the same rates or variation exists when extended to all patients and payors.

Accordingly, we sought to assess the rate of unplanned hospital visits among patients undergoing ambulatory surgery in the United States. Specifically, we sought to assess the rate, timing, and reasons for unplanned hospitals visits within the first 30 days following same-day surgery at either HOPDs or ASCs and how these differed across different types of same-day surgeries. Lastly, we assessed how the risk-adjusted rate of hospital visits varied among outpatient surgical facilities.

## Methods

We used state-level, administrative data from the Agency for Healthcare Research and Quality's (AHRQ) Healthcare Cost and Utilization Project [20]. Specifically, California (CA) data on ambulatory surgery (State Ambulatory Surgery and Services Databases (SASD)) [21], inpatient (State Inpatient Databases (SID)) [22] and emergency department (State Emergency Department Databases (SEDD)) [23] utilization was used from 2009–2011, the most recent period of time available that spans more than 12 months [24]. California was selected for analysis because its database contains unique variables that allow patients to be followed over time and across the ambulatory surgery, inpatient, and emergency department settings, the quality of its data, and its large population. These data are a census of discharges from free-standing and hospital-affiliated ambulatory surgery centers. Data does not include physician-owned free-standing surgery centers, as the State of California does not require physician-owned surgery clinics to obtain a facility license, and so are not required by law to report data. Information for each patient discharge includes up to 21 Current Procedural Terminology or International Classification of Diseases, Ninth Revision, Clinical Modification (ICD-9-CM) procedures codes, 15 diagnostic ICD-9-CM codes, and information about patient demographics, anticipated payer and discharge disposition.

## Study cohort

We included all individuals who underwent same-day surgeries performed in HOPDs and ASCs between January 1, 2009-November 30, 2011 who were at least 18 years of age and had valid, encrypted, person-level identifiers. Next, we sequentially excluded discharges where the disposition was listed as missing, death, or left against medical advice. We limited this analysis to facilities that performed at least 100 surgeries to obtain a reliable estimate of the hospital visit rate.

We identified same-day surgeries based on the list of covered ambulatory surgery center (ASC) procedures released by Medicare to identify surgeries that can be safely performed as same-day surgeries and do not typically require an overnight stay [25]. Surgeries on the ASC list of covered procedures do not involve or require: major or prolonged invasion of body cavities; extensive blood loss; major blood vessels; or care that is either emergent or life-threatening. These procedures typically require less than 90 min operating time with a <4-6-hour post-operative recovery period. While this list of procedures was developed specifically for ASCs in the Medicare population, the reasoning appears to be appropriate for same-day surgery despite location or age, so we used the same list to identify same-day surgeries also performed in HOPDs in the HCUP database. We were primarily interested in outcomes of major surgical procedures. Accordingly, we only include surgeries considered "substantive surgeries" by Medicare, as identified by the global surgical indicator code 090. We did not include minor surgeries or procedures (global indicator codes 000 and 010) which typically include non-operative procedures (e.g. application of cast), simple diagnostics tests such as biopsies, endoscopic procedures, and other minor surgical procedures.

Procedures were grouped using the Agency for Healthcare Quality and Research (AHRQ) multi-level procedural Clinical Classification System (CCS) [26]. The procedural CCS allows aggregation of procedure groups into anatomical body systems such as Eye, Urinary, and Gastrointestinal surgery which aligns with surgical specialities in clinical practice.

## Main outcome variables

Our primary outcome was the rate of all-cause, unplanned hospital visits within 30 days of the same-day surgery, defined as (1) any direct transfer for an inpatient admission after surgery or (2) any emergency department visit, observation stay visit, or unplanned inpatient admission occurring within 30 days after the patient is discharge from the HOPD or ASC.

We included only unplanned admissions because planned admissions typically do not represent adverse events. We did not include hospital admissions that were admitted directly (ie; without a previous emergency department visit) if the primary diagnosis was maintenance radiation or chemotherapy, rehabilitation services, cancer, or normal obstetrical delivery (S1 Appendix). We considered all inpatient admissions occurring on the day of surgery (day 0) for ASCs as unplanned, and therefore included direct hospital transfers as unplanned admissions. For all other inpatient admissions, occurring after Day 1 for HOPDs, we excluded 'planned' admissions. For all hospital-based, acute care encounters, we recorded the primary diagnostic categories associated with the encounter based on the AHRQ clinical classification groupings of ICD-9-CM codes [26].

## Descriptive variables

Patient characteristics were obtained for both descriptive and risk-standardization purposes, including patient age, sex, race and ethnicity as defined in HCUP (White, Black, Hispanic, other and missing), primary payer (Medicare, Medicaid, private, other), and the first listed procedure associated with the discharge when grouped by the AHRQ classification of *Current Procedural Terminology* coding. We assessed comorbidity according to the enhanced-Elixhauser algorithm described previously [27,28], which identifies 31 chronic medical conditions. A patient was considered to have a condition if it was a listed diagnosis during the initial discharge from the ambulatory surgery center or any discharge in the previous 6 months from a hospital admission or emergency department.

## Statistical analysis

Data is summarized as frequencies and percentages for categorical variables. Continuous variables are presented as mean +/- standard deviation. The Chi-square test, student's t-test or Wilcoxon ranked sum tests were used to compare characteristics between groups as appropriate.

We calculated the observed rate of hospital visits within 30 days of the same-day surgery. This rate was calculated by dividing the number of events (numerator) by the number of same-day procedures (denominator) expressed as the rate per 100 procedures. Descriptive statistics were used to estimate average rates with 95% confidence intervals (95% CI) for the overall sample and the 20 most frequently performed procedures.

To compare rates across centers, we estimated the age, sex, and procedure-adjusted rates of hospital visits for each ASC or HOPD using 2-level, hierarchical, generalized linear mixed models and methods similar to existing Centers for Medicare and Medicaid Services measures [29,30]. A single model was used to estimate the log odds of an unplanned hospital visit conditional on age, sex and procedure assuming a binomial distribution and logit link [31]. The first level included fixed effects for patient age, sex, and procedure type (a 21-level categorical

variable specifying the 20 most common procedures by volume or "other"), whereas the second level included an ASC/HOPD random intercept. For each ASC/HOPD, the rate was calculated as the ratio of the number of "predicted" outcomes (obtained from a model applying the hospital-specific effect) to the number of "expected" outcomes (obtained from a model applying the average effect among hospitals), multiplied by the unadjusted rate for the entire sample. For this analysis, we limited the sample to ASC/HOPDs reporting at least 100 procedures (>5th percentile by volume in the overall sample) meeting the above criteria during the study period to avoid unstable parameter estimates.

Finally, we compared the 30-day hospital visit rate among outpatient facilities adjusting for patient characteristics and type of surgery performed at the facility (using surgery body system grouping) accounting for clustering of patient and surgical volume at each facility using the same method described above. Facility variation was measured overall and separately for HOPDs and ASCs.

All analysis was conducted using SAS version 9.2 (SAS Institute, Cary, NC). Because this study used publicly available data that do not include patient identifiers, our study was considered exempt from review by the Yale University Human Investigations Review Board and approved through a data use agreement with HCUP.

## Results

A total of 1,260,619 ambulatory same-day surgeries from 440 surgical facilities in the state of California met our study inclusion criteria. Among these surgeries, 132,051 (10.5%) were performed at 56 ambulatory surgical centers and the remaining 1,128,568 (89.5%) were performed at 384 hospital outpatient departments.

### Patient characteristics

The mean age of patients having same-day surgery was 52.1 years and 43.7% of patients were male (Table 1). Of all same-day surgeries, 67.3% of patients were White, 20.9% of patients labeled Hispanic, 5.7% 'Asian or Pacific' and 4.2% of patients African American, while 21.1% had race missing in the dataset. Comorbidities were infrequent among same-day surgical patients with 10% having more than two recorded comorbidities. The distribution of age (mean age 50.6 vs 52.2 years, p = 0.79), sex (52 vs 51% female, p = 0.23) and race were similar between ASCs and HOPDs. However, important comorbidities such as heart conditions (e.g., congenital heart disease, cardiac arrhythmia, valvular disease: (6.96% vs. 0.14%), diabetes

**Table 1. Characteristics of patients undergoing ambulatory same-day surgery at HOPDs and ASCs.**

| Variable | Total | ASC # (%) | HOPD # (%) |
|---|---|---|---|
| Patients (N) | 1,260,619 | 132,051 (10.5) | 1,128,568 (89.5) |
| Facilities (N) | 440 | 56 | 384 |
| Demographic Characteristics | | | |
| Age in years, mean | 52.1 (16.8) | 50.6 (15.4) | 52.2 (17.0) |
| Female | 649,335 (51.5) | 69,195 (52.4) | 580,140 (51.4) |
| Medicare Recipient | 260,398 (20.7) | 18,885 (14.3) | 241,513 (21.4) |
| Elixhauser Score | | | |
| 0 | 712,185 (56.5) | 127,957 (96.9) | 584,228 (51.8) |
| 1–2 | 421,829 (33.5) | 4,093 (3.1) | 417,736 (37) |
| 3–4 | 108,901 (8.6) | 264 (0.2) | 108,637 (9.6) |
| 5+ | 17,704 (1.4) | 132 (0.1) | 17,572 (1.6) |

(10.3% vs. 0.15%), chronic lung disease (8.2% vs. 0.04%) and renal disease (4.5% vs. 0.14%) were significantly more frequent among patients having surgery at HOPDS, with 7.1% of these patients having 3 or more comorbidities compared with 0.3% for ASCs (p<0.001).

## Surgery characteristics

When grouped by anatomical body system, "muscular-skeletal" type of surgery (33.8%) was the most frequent same-day surgery performed overall and among Medicare beneficiaries (26.1%) (Table 2). Digestive system (18.7%) and Integumentary system (11.3%) surgeries were the next most frequent procedures performed as same-day surgery. While most surgeries were performed both in ASCs and HOPDs, the volume of case type was different depending on organ system. For example, Digestive (19.8% of HOPD cases vs. 9.6% of ASC cases), Cardio-vascular (6.1% vs. 1.1%), Urinary (7.7% vs. 4.2%), Male Genitalia (2.4% vs 1.6%), and Female Genitalia (8.3% vs. 3.4%) surgery made up a greater proportion of HOPDs surgeries compared with ASCs. In contrast, surgery performed on the Musculoskeletal system (30.9% of HOPD cases vs. 58.2% of ASC cases) and Nervous System (3.3% vs. 6.4%) made up a greater propor-tion of procedures performed at ASCs.

## Outcomes

Among all of the included patients undergoing same day surgery at an ASC or HOPD, most unplanned hospital visits within 30 days were post-discharge hospital visits with fewer than 10% representing same-day admission (Fig 1). Several patient characteristics were associated with increased risk of unplanned hospitals visits. Specifically, in bivariate analyses, increasing age (by 10-year increments) was associated with an odds ratio (OR) of 1.02 (95% CI: 1.01–1.02), female sex (OR 1.02, 95% CI: 1.00–1.04), comorbidities (based on higher Elixhauser Score), type of procedure (p<0.001) and the procedure being done in an HOPD (OR 1.37, 95% QI: 1.21–1.54) (Table 2).

Overall, the body system with the highest surgical volume was musculoskeletal, followed by the digestive system. This was the same in ASCs and HOPDs. Overall, the body system with the highest mean 30-day observed outcome rate was cardiovascular (8.3%) followed by urinary (6.2%) (Fig 2). For ASCs, the body system with the highest 30-day observed outcome rate was split between urinary and cardiovascular (4% each) and for HOPDs was 8.4% for cardiovascu-lar followed by 6.3% for urinary. The body system procedures with the lowest observed out-come rates were ear (1.9%), endocrine (2.2%), nervous system (2.2%) and skin and breast (2.2%). These rates were even lower for ASCs with nervous system being the lowest (1.4%)

**Table 2. Bivariate predictors of 30-day unplanned hospital visits.**

| Variable | Odds Ratio | 95% Confidence Interval | p-value |
|---|---|---|---|
| Age (10-year increments) | 1.02 | (1.01–1.02) | p<0.01 |
| Female | 1.02 | (1.00–1.04) | p = 0.04 |
| Elixhauser Score | | | |
| 0 | Reference | | |
| 1–2 | 1.35 | (1.32–1.38) | |
| 3–4 | 1.98 | (1.92–2.05) | |
| 5+ | 2.94 | (2.78–3.10) | |
| Type of Facility | | | |
| ASC | Reference | | |
| HOPD | 1.37 | (1.21–1.54) | p<0.01 |

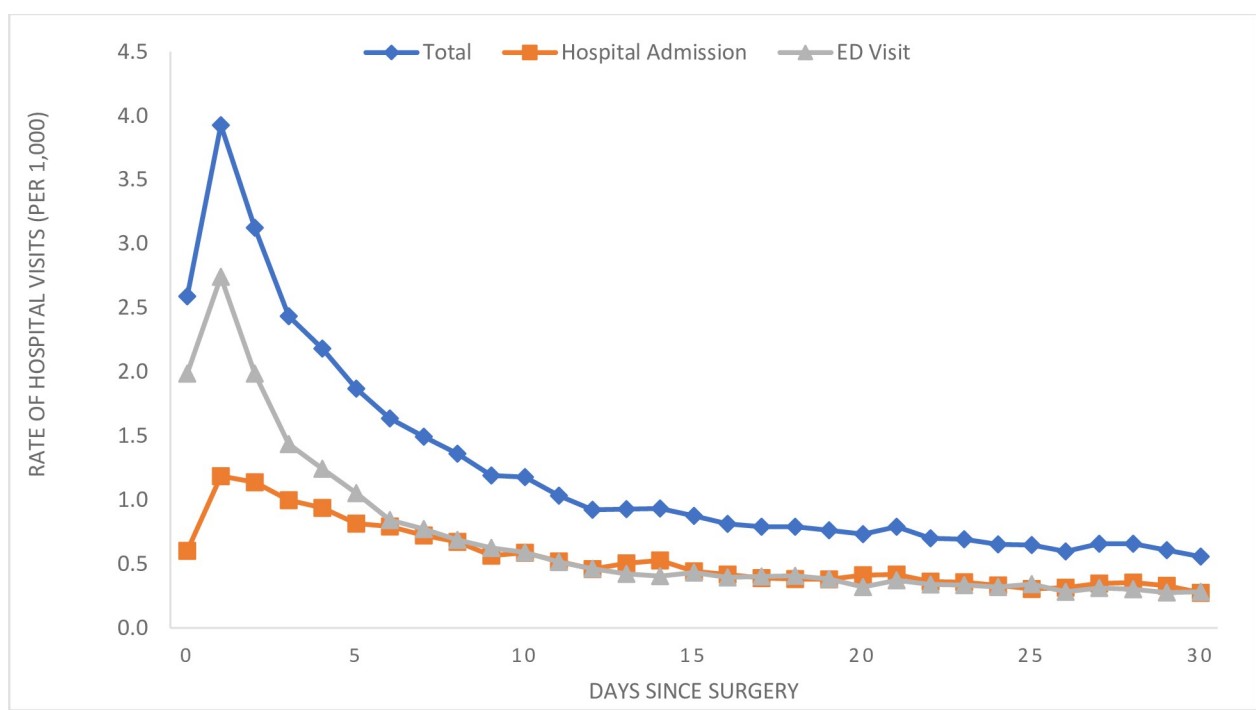

**Fig 1. Daily rate of unplanned hospital visits following all same day surgery.**

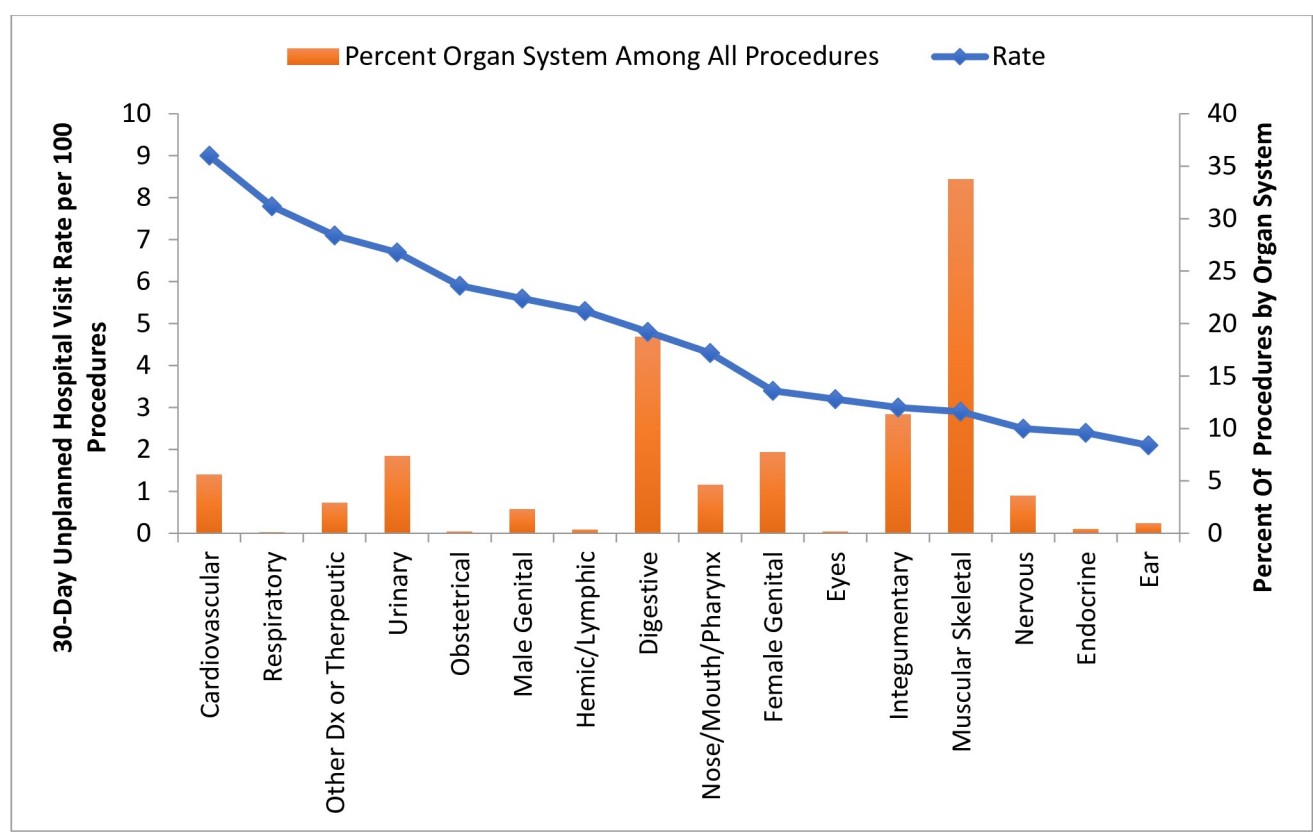

**Fig 2. Overall rate of unplanned hospital visits by system.**

**Table 3. Risk adjusted rates by organ system.**

| | Mean Risk Adjusted Rates by Organ System | | |
| --- | --- | --- | --- |
| | %, (95% Confidence Interval) | | |
| Body System | ASC | HOPD | Total |
| Eye | 3.7 (1.6, 8.1) | 3.7 (2.8, 4.9) | 3.7 (2.4, 5.6) |
| Musculoskeletal | 2.6 (2.3, 2.9) | 3.7 (3.6, 3.9) | 3.1 (3.0, 3.3) |
| Urinary | 5.7 (4.7, 6.8) | 7.6 (7.2, 8.0) | 6.6 (6.0, 7.2) |
| Nervous | 1.8 (1.5, 2.3) | 3.0 (2.8, 3.3) | 2.4 (2.1, 2.7) |
| Endocrine | 0.2 (0.01, 0.7) | 2.7 (2.2, 3.4) | 0.8 (0.03, 0.19) |
| Ear | 4.1 (2.5, 6.7) | 2.5 (2.1, 2.9) | 3.2 (2.5, 4.2) |
| Nose/Mouth/Pharynx | 3.7 (3.0, 4.5) | 5.8 (5.5, 6.2) | 4.7 (4.3, 5.2) |
| Respiratory | 7.0 (1.1, 33.6) | 8.1 (6.5, 10.2) | 7.6 (3.0, 17.6) |
| Cardiovascular | 5.2 (3.8, 6.9) | 7.2 (6.8, 7.6) | 6.1 (5.3, 7.0) |
| Lymph | 2.5 (0.8, 7.7) | 5.6 (4.8, 6.4) | 3.8 (2.1, 6.6) |
| Digestive | 4.0 (3.4, 4.6) | 6.0 (5.7, 6.3) | 4.9 (4.5, 5.3) |
| Male Genital | 3.1 (2.2, 4.3) | 6.7 (6.3, 7.2) | 4.6 (3.9,5.4) |
| Female Genital | 3.9 (3.1, 4.9) | 4.6 (4.3, 4.8) | 4.2 (3.7, 4.7) |
| Obstetric | 0.2 (NE) | 9.7 (8.0, 11.8) | 1.6 (NE) |
| Integumentary | 2.7 (2.2, 3.2) | 3.4 (3.2, 3.6) | 3.0 (2.8, 3.3) |
| Other Diagnostic/Therapeutic | 1.0 (0.2, 6.4) | 6.3 (6.0, 6.8) | 2.6 (1.0, 6.4) |
| Total Across Organ Systems | 3.5 (3.2, 3.9) | 5.0 (4.7, 5.4) | 4.3 (4.0, 4.7) |

NE: Non-estimable.

Adjusted predictors: Gender, age, Elixhauser Index.

followed by skin and breast and musculoskeletal (both 1.7%). For HOPDs, the lowest observed outcome rates were in procedures among the ear body system (1.9%), followed by endocrine (2.2%) and nervous system (2.5%).

We then calculated risk-adjusted rates overall and by system (Table 3). Overall, the risk adjusted 30-day rate of unplanned hospital visits was 4.8%, with emergency department visits of 3.1% and hospital admissions of 1.7%. Most of these admissions were post-discharge unplanned hospital visits. The rate of same-day admission was very low at 0.28%. However, the mean risk-standardized rate was significantly lower among ASCs (3.5%, 95% CI, 3.2–3.9) when compared with HOPDs (5.0%, 95%CI, 4.7–5.4).

Importantly, we also looked at variation among facilities as a potential marker of quality as variability could represent the ability to improve care. When we looked at overall facility variation, we found that it was significant with an estimated variance of 0.21 (p<0.0001) suggesting a high level of variation among facilities (Fig 3).

## Discussion

In this study, we sought to assess the rate of unplanned hospital visits among patients undergoing ambulatory surgery in the state of California including the rate, timing, and reasons for unplanned hospitals visits within the first 30 days following same-day surgery at either HOPDs or ASCs and how these differed across different types of same-day surgeries. We found that rates of unplanned hospital visits vary greatly depending on type of procedure, as well as type of facility. Additionally, we aimed to assess how the risk-adjusted rate of hospital visits varied among outpatient surgical facilities. We found this variation to be high among facilities, suggesting possible variation in quality of care.

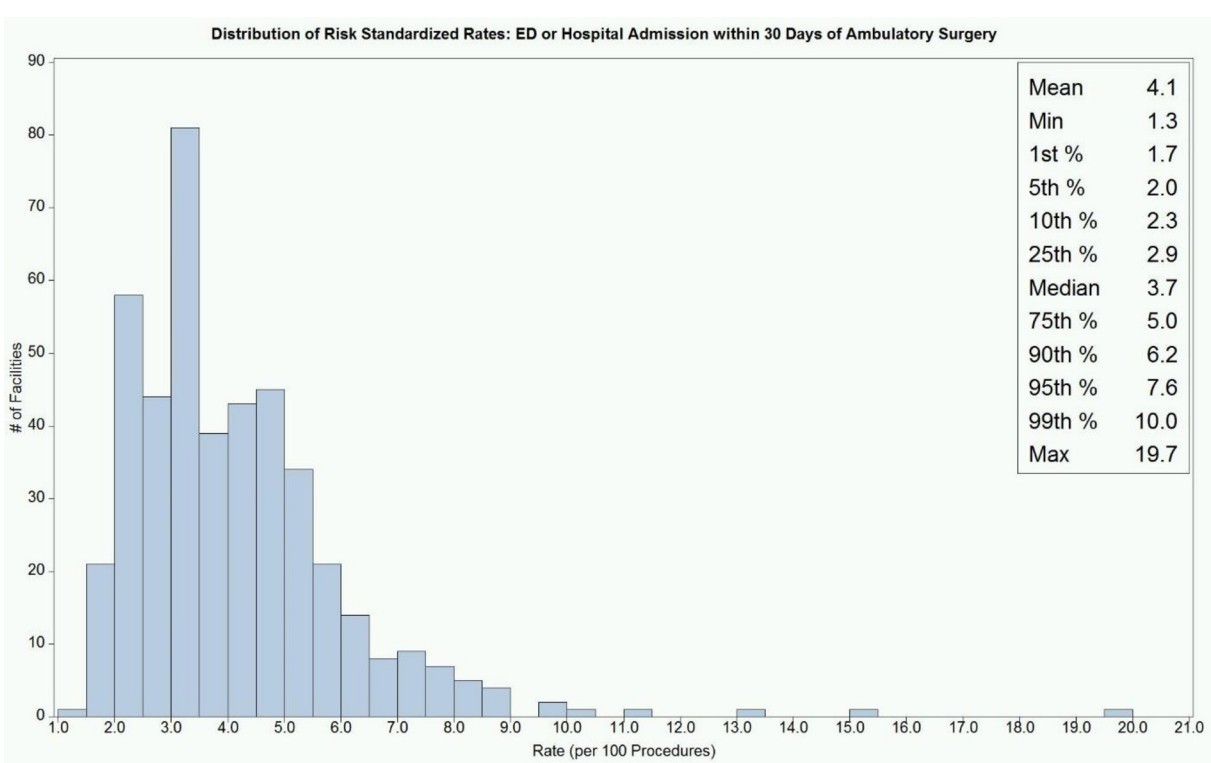

**Fig 3. Risk standardized rates, all organ systems.** Adjusted predictors: Gender, age, Elixhauser Index, organ system, type of facility.

The current study supports other reports identifying variation in same day surgery [18,19]. However, most of these reports are limited in that they focus on only one type of procedure or in one specialty. In this study, we analyze both ASCs and HOPDs, as well as all major procedures performed as same day surgery. In this large cohort, we still found risk adjusted rates of unplanned hospital visits of 4.8% which is not insignificant in what is otherwise a healthy population and may be preventable given many of the hospital visits were within the first few days. Given the early occurrence of the majority of unplanned hospital visits, early follow-up may be beneficial. Finally, though the vast majority of these visits occur early, unplanned hospital visits do occur over the full 30-day period. It is important to note that most of these cases are performed in HOPDs, making these facilities an important focus for quality improvement metrics.

Additionally, we found important predictors of interest which may allow us to focus any quality interventions on specific 'high-risk' groups of patients. Specifically, we found that older patients were more likely to have an unplanned hospital visit, as well as women and patients who had a same day surgery in an HOPD. This may be because patients who are sicker will be preferentially cared for in an HOPD though this needs to be further studied. Additionally, we found specific procedures that had higher rates, specifically cardiovascular and urinary procedures which are commonly performed but have high rates of unplanned hospital visits. Aggregating these risk predictors together may allow us to define cohorts of patients who could be a target for more targeted follow-up care after discharge from a same-day surgery.

Our study had several limitations. First, our data is not recent. However, to date it represents the most recent multiple years in a row of California data that exist. As the trend towards same day surgery has only increased, and to date there are no new outcomes being tracked or

required quality reporting on these facilities. Importantly, we were unable to adjust for race as greater than 20% of the same day surgery visits were missing race. Therefore, we are unable to account for racial disparities in the outcomes. As our population ages, it is possible that more older adults are undergoing same-day surgery, the group representing the highest risk for an unplanned hospital visit. Finally, we report only on data from the State of California. California was selected for analysis because its database allows patients to be followed over time and across the ambulatory surgery, inpatient, and emergency department settings, is of good quality, and has the large population in any single state in the country.

The findings of this study may have broad implications in quality control for ambulatory surgery. We found considerable variation in 30-day unplanned hospital visit rates among ambulatory surgical facilities among different types of procedures and between ACSs and HOPDs. This suggests that surgical quality or care processes vary substantially among outpatient facilities and may suggest an area of great improvement for patient care. Further, a high rate of unplanned hospital visits, especially if these visits are for common and preventable reasons, may suggest considerable opportunity to improve the quality care among ambulatory surgical facilities in the United States. This also suggests that the CMS quality measure may be generalizable to an all payor population. Further research is needed to better understand what types of problems patients are having leading to unplanned hospital visits, and to create systems to mitigate these issues including transparent tracking and reporting of data.

## Supporting information

**S1 Appendix. ICD-9-CM coding appendix.** *ICD-9-CM*, International Classification of Diseases, Ninth Revision, Clinical Modifications.
(DOCX)

## Acknowledgments

We would like to acknowledge the National Clinician Scholars Program and everyone who supports the program at Yale University as well as other sites across the country. We would also like to acknowledge Veronika Shabanova for her skilled biostatistical support throughout the project.

## Author Contributions

**Conceptualization:** Tasce Bongiovanni, Craig Parzynski, Isuru Ranasinghe, Joseph S. Ross.

**Data curation:** Tasce Bongiovanni, Isuru Ranasinghe.

**Formal analysis:** Tasce Bongiovanni, Craig Parzynski, Joseph S. Ross.

**Investigation:** Tasce Bongiovanni.

**Methodology:** Tasce Bongiovanni, Craig Parzynski, Isuru Ranasinghe, Michael A. Steinman, Joseph S. Ross.

**Project administration:** Tasce Bongiovanni, Isuru Ranasinghe, Joseph S. Ross.

**Resources:** Joseph S. Ross.

**Software:** Craig Parzynski.

**Supervision:** Michael A. Steinman, Joseph S. Ross.

**Validation:** Craig Parzynski, Isuru Ranasinghe, Michael A. Steinman.

**Visualization:** Tasce Bongiovanni, Michael A. Steinman.

**Writing – original draft:** Tasce Bongiovanni.

**Writing – review & editing:** Tasce Bongiovanni, Craig Parzynski, Isuru Ranasinghe, Michael A. Steinman, Joseph S. Ross.

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
