## [Decision Letter · Decision Letter 0]

18 Jun 2021

Unplanned Hospital Visits after Ambulatory Surgical Care

PONE-D-21-13306

Dear Dr. Bongiovanni,

We’re pleased to inform you that your manuscript has been judged scientifically suitable for publication and will be formally accepted for publication once it meets all outstanding technical requirements.

Kind regards,

Hitesh Raheja, MD

Academic Editor

PLOS ONE

1. We noted in your submission details that a portion of your manuscript may have been presented or published elsewhere. [Yes as a preprint. This manuscript was previously published in medRxiv: https://doi.org/10.1101/2021.03.10.21249910] Please clarify whether this [conference proceeding or publication] was peer-reviewed and formally published. If this work was previously peer-reviewed and published, in the cover letter please provide the reason that this work does not constitute dual publication and should be included in the current manuscript.

2. Please respond by return e-mail with an updated version of your manuscript to include your abstract after the title page.

Reviewers' comments:

Reviewer's Responses to Questions

**Comments to the Author**

1. Is the manuscript technically sound, and do the data support the conclusions?

Reviewer #1: Yes

Reviewer #2: Yes

2. Has the statistical analysis been performed appropriately and rigorously? 

Reviewer #1: Yes

Reviewer #2: Yes

3. Have the authors made all data underlying the findings in their manuscript fully available?

Reviewer #1: Yes

Reviewer #2: Yes

4. Is the manuscript presented in an intelligible fashion and written in standard English?

Reviewer #1: Yes

Reviewer #2: Yes

5. Review Comments to the Author

Reviewer #1: well written article

Gives a great overview on all the undefined risks for ambulatory surgical care

Gives a overview of different surgical procedures, prior research articles have been limited to only a certain set of procedures.

Provides scope for improvement of patient care.

Reviewer #2: I think it is good to assess the rate of unplanned hospital visits among patients

undergoing ambulatory surgery in the United States as it will improve the overall quality and safety among patients undergoing short surgery care and who are at risk for more preventable harms and how these events can be controlled or monitored.

6. PLOS authors have the option to publish the peer review history of their article (what does this mean?). If published, this will include your full peer review and any attached files.

Reviewer #1: **Yes: **Deepti Bhandare

Reviewer #2: **Yes: **Bhunesh Maheshwari, MD

---

## [Editor Report · Acceptance letter]

8 Jul 2021

PONE-D-21-13306 

Unplanned Hospital Visits after Ambulatory Surgical Care 

Dear Dr. Bongiovanni:

I'm pleased to inform you that your manuscript has been deemed suitable for publication in PLOS ONE. Congratulations! Your manuscript is now with our production department. 

Kind regards, 

on behalf of

Dr. Hitesh Raheja 

Academic Editor

PLOS ONE